# Molecular and Morphological Evidence Reveals Four New *Neocosmospora* Species from Dragon Trees in Yunnan Province, China

**DOI:** 10.3390/jof11080571

**Published:** 2025-07-31

**Authors:** Mei Jia, Qi Fan, Zu-Shun Yang, Yuan-Bing Wang, Xing-Hong Wang, Wen-Bo Zeng

**Affiliations:** 1Yunnan Microbiological Institute, School of Life Science, Yunnan University, Kunming 650091, China; jiamei1027@mail.ynu.edu.cn; 2Key Laboratory of Phytochemistry and Natural Medicines, Kunming Institute of Botany, Chinese Academy of Sciences, Kunming 650201, China; fanqi_666@163.com (Q.F.); wangyuanbing@mail.kib.ac.cn (Y.-B.W.); 3Yunnan Key Laboratory for Fungal Diversity and Green Development, Kunming Institute of Botany, Chinese Academy of Sciences, Kunming 650201, China; 4Yunnan Center for Disease Control and Prevention, Kunming 650022, China; yzsylc@126.com; 5Yunnan Key Laboratory of Fermented Vegetables, Honghe 661100, China; 6Sanqi Medicines College, Wenshan University, Wenshan 663099, China

**Keywords:** *Neocosmospora*, *Dracaena*, new species, taxonomy, phylogeny

## Abstract

*Neocosmospora* (*Nectriaceae*) is a globally distributed fungal genus, traditionally recognized as a group of plant pathogens, with most members known to cause severe plant diseases. However, recent studies have demonstrated that many of these fungi can also colonize plants endophytically, with certain strains capable of promoting plant growth and stimulating the production of secondary metabolites. In this study, 13 strains of *Neocosmospora* were isolated from the stems and leaves of *Dracaena cambodiana* and *D. lourei* in Yunnan Province, China. To clarify the taxonomic placement of these strains, morphological examination and multi-gene (ITS, nrLSU, *tef1*, *rpb1*, and *rpb2*) phylogenetic analyses were performed. Based on morphological and phylogenetic evidence, four new species are introduced and described here: *N. hypertrophia*, *N. kunmingense*, *N. rugosa*, and *N. simplicillium*. This study expands our understanding of the fungal diversity associated with *Dracaena*, provides essential data for the taxonomy of *Neocosmospora*, and serves as a resource for the future development and utilization of *Neocosmospora* endophytes.

## 1. Introduction

*Dracaena* (dragon trees) is native to the tropical regions of Africa and Asia and comprises over one hundred species. Among them, *D. cochinchinensis*, *D. cambodiana*, *D. loureiri*, *D. draco*, and *D. cinnabari* can produce draconis sanguis (dragon’s blood), which is commonly known as “Xuejie” in China and has been used as a traditional medicine for over 1500 years. Due to the slow growth of dragon trees and the high demand for dragon’s blood, wild *Dracaena* resources have been severely overexploited. Consequently, numerous *Dracaena* species, including *D. draco* and *D. cinnabari*, have been listed on the International Union for the Conservation of Nature and Natural Resources (IUCN) Red List of Threatened Species (www.iucnredlist.org) [1].

Due to the frequent occurrence of fungi in the stems of dragon trees that produce draconis sanguis, researchers have hypothesized a potential association between fungi and the formation of dragon’s blood. In 2010, over 300 fungal strains were isolated by Jiang et al. [2] from dragon trees producing dragon’s blood, and the predominant isolates were identified as fusarioid fungi. Experimental results have demonstrated that *F. proliferatum* can induce dragon’s blood production in *D. cochinchinensis* [2,3,4,5]. Investigating the fusarioid fungi associated with *Dracaena* can facilitate the selection of fungal strains that effectively induce draconis sanguis production. This study may enable the controlled induction of dragon’s blood production to meet the current demand and protect wild dragon tree populations and biodiversity.

*Neocosmospora*, a genus within *Nectriaceae*, is phylogenetically allied to *Fusarium*. The genus was established by Smith in 1899 based on ascospore morphology, with *N. vasinfecta* designated as the type species. Morphologically, *Neocosmospora* species typically have yellow, orange, or red-brown macroconidia with thick-walled, blunt, and rounded apical cells and inconspicuous foot-shaped basal cells; microconidia phialides are very long and narrow [6]. Species of *Neocosmospora* are also globally distributed and commonly occur in soil, plant debris, living plant tissues, air, and water [7]. They are known to parasitize over 500 plant species across more than 100 families, causing a wide range of plant diseases, including dry and jelly end potato rot, head blight of wheat (*Triticum aestivum*), and root rot of *Citrus* spp., pea, peanut, sweet potato, and wheat [8,9,10,11,12,13]. These fungi are also known to cause keratitis, skin infections, and other opportunistic infections in humans and animals [7,14]. However, some endophytic species within this genus play important roles in agriculture and medicine due to their potent bioactivity. For instance, endophytic *N. solani* can produce naphthoquinones and aza-anthraquinones with cytotoxic and antibacterial activities, immunologically active polysaccharides, and antioxidant flavonoids [15,16,17,18]. Furthermore, certain *Neocosmospora* species have demonstrated biocontrol potential against phytopathogenic fungi and parasites in agricultural settings [19,20,21]. Currently, there are more than 140 accepted species in this genus [6,22,23,24,25,26].

In this study, 13 fungal isolates were obtained from *Dracaena* samples collected in Kunming and Pu’er Cities, Yunnan Province, China. To clarify their taxonomic placement and phylogenetic relationships, species identification was performed using an integrated approach that combined multi-gene phylogenetic analyses (ITS, nrLSU, *tef1*, *rpb1*, and *rpb2*) and morphological observations. The results not only enhance our understanding of the fungal diversity associated with *Dracaena* but also provide a scientific basis for exploring potential microbial resources for the artificial induction of dragon’s blood production.

## 2. Materials and Methods

### 2.1. Sample Collection and Strains Isolation

Specimens of *D. cambodiana* and *D. lourei* were collected from Kunming and Pu’er Cities in Yunnan Province between September 2019 and December 2020. One tree was sampled every 50 km to ensure the geographic coverage. Trunks, leaves, taproots, and large lateral roots were collected from trees with stem diameters of 3–5 cm to minimize the developmental variation. All samples that were collected from asymptomatic trees were stored in sterile bags and transported to the laboratory within 48 h. Leaves were cut into 3–4 cm segments from different canopy levels, and stems and roots were cut into 4–5 cm pieces. Surface sterilization included 1.5% NaOCl (2 min), 70% ethanol (2 min), rinsing three times with sterile water, and drying [27]. The segments were further cut into smaller pieces before placement on PDA supplemented with streptomycin sulfate (0.5 g/L) and penicillin (0.4 g/L). Five to six fragments were placed on each plate and incubated at 25 °C.

After 2–8 days of incubation, colonies exhibiting morphological characteristics consistent with *Neocosmospora* were subcultured on fresh media. To prevent bacterial contamination, all media were supplemented with streptomycin sulfate (0.5 g/L) and penicillin (0.4 g/L). Bacterial contamination was managed through repeated purification and single-spore isolation. All strains were stored at the Kunming Institute of Botany, and Living cultures were deposited in the Culture Collection of Kunming Institute of Botany, Chinese Academy of Sciences (KUNCC), and Yunnan Institute of Microbiology (YIM). Voucher specimens (from dried cultures) were deposited in the Herbarium of Cryptogams at the Kunming Institute of Botany, Chinese Academy of Sciences (HKAS), China.

### 2.2. DNA Extraction, PCR Amplification, and Sequencing

Total genomic DNA was extracted from fresh mycelia grown on PDA using a modified CTAB protocol [28]. Five genes, including the 5.8S nuclear ribosomal RNA gene with two flanking internal transcribed spacer (ITS) regions, partial translation elongation factor (*tef1*), partial RNA polymerase largest subunit (*rpb1*), partial RNA polymerase second largest subunit (*rpb2*) gene regions, and 28S large subunit of the nrDNA (nrLSU), were amplified and sequenced. All primers and PCR amplification procedures were performed as described in previous studies [29,30,31,32,33,34] and are listed in Table 1. Consensus sequences for each marker were assembled using MEGA 7 [35]. All newly generated sequences were submitted to GenBank.

### 2.3. Phylogenetic Analyses

Species of *Neocosmospora* were searched based on the latest published articles and Index Fungorum to collect DNA sequences representing the phylogenetic diversity of the genus. Reference sequences of *Neocosmospora* species were downloaded from NCBI. The GenBank accession numbers are listed in Appendix A. A total of 123 species of gene sequence data were collected, including four new species described in this study and the outgroup (*Geejayessia atrofusca* and *G. cicatricum*). Sequence alignment was performed for each gene region using MAFFT V.7 and manually checked and corrected using MEGA V.7, when necessary [35,36]. Ambiguous alignment sites were excluded from the phylogenetic analysis, and gaps were considered as missing data. After sequence alignment, the sequences of the five genes were concatenated. Phylogenetic analyses of the five genes were conducted using the Maximum Likelihood (ML) and Bayesian Inference (BI) methods. ModelFinder was used to estimate the best-fit evolution model for each gene in IQ-Tree. ML analysis was performed using RAxML v7.9.1 with 1000 rapid bootstrap replicates on the five genes [37]. The Bayesian analysis was performed using MrBayes V. 3.2.6 and consisted of four 90-m generations running in parallel, starting from a random tree topology with sampling frequencies per 1000 generations [38,39]. The 50% majority rule consensus tree and posterior probability (PP) values were calculated after discarding the initial 25% of the saved trees as the “burn-in phase.” Each gene phylogenetic examination significantly supported clade conflict with PP > 0.9 and BS > 70% concatenated for the five genes datasets [40]. The phylogenetic tree was visualized and modified using the Chiplot online tool (https://www.chiplot.online/) [41].

### 2.4. Genealogical Concordance Phylogenetic Species Recognition Analysis

The pairwise homology index (Φw) test, based on the Genealogical Concordance Phylogenetic Species Recognition (GCPSR) principle, was employed to define phylogenetically related, ambiguous species. The pairwise homoplasy index (PHI) test was performed using Splitstree v. 6.3.27 to determine the extent of recombination within phylogenetically closely related species using a five-gene concatenated dataset [42,43]. The relationships among these four closely related species groups were visualized by constructing split graphs from the five-gene concatenated datasets, employing both the LogDet transformation and split decomposition options. If the pairwise homogeneity index results were below 0.05 (*p* < 0.05), this would indicate significant recombination within the dataset.

### 2.5. Morphological Observation

A small number of mycelia were placed on medium (carnation leaf agar(CLA), synthetic low-nutrient agar(SNA), and potato dextrose agar (PDA)) with a diameter of 5 mm and covered with a cover slide for microscope slide culture under a 12/12 h near-ultraviolet light/dark cycle for 7–14 d at 25 °C [44,45]. The general micromorphological characteristics of asexual forms (microconidia, macroconidia, chlamydospores, and conidiophores) were observed and measured using Olympus BX53 microscopes (Tokyo, Japan). For a more detailed description of the asexual morph, sporodochia formed on the surface of carnation leaves were photographed using an Olympus SZ61 stereomicroscope. To assess the growth rate, agar blocks of approximately 5 mm in diameter were removed from the colonies and placed on new PDA plates and OA. Colonies were grown at 25 °C in the dark and photographed and measured on the seventh day. The color, shape, and odor of the colonies on the medium were also recorded. At least 30 randomly selected elements were recorded for each fungal structure.

## 3. Results

### 3.1. Multi-Gene Phylogeny

In this study, 13 strains of novel *Neocosmospora* species were isolated from the roots, stems, and leaves of *D. cambodiana* and *D. lourei* in Yunnan Province. Phylogenetic analysis grouped these strains into four distinct species-level clades within *Neocosmospora*, whrerin four species exhibited significant genetic divergence and formed independent clades with strong support. These findings indicated that they represent putative novel species.

In the ML and BI phylogenetic analyses, the combined five-gene dataset of 123 species selected in this study was employed to construct the phylogenetic framework of the genus *Neocosmospora*. Two taxa of *Geejayessia* (*G. atrofusca* NRRL 22316 and *G. cicatricum* CBS 125552) were designated as outgroups. The five-gene dataset contained 4268 characters (*tef1* 662, ITS 475, nrLSU 483, *rpb1* 1390, and *rpb2* 1618). The best-fit evolutionary models selected according to the Akaike criterion were GTR+F+R3 for *tef1* and ITS, TIM2e+R2 for nrLSU, TIM3+F+I+G4 for *rpb1*, and GTR+F+R5 for *rpb2*. The topology resulting from both ML and BI analyses was consistent (Figure 1). Five-gene analyses revealed the presence of 123 species-level clades within this taxon, distributed among the four primary clades, consistent with the findings of Sandoval-Denis [6]. The four new species formed distinct phylogenetic clades, each supported by high BS and PP values (*N. hypertrophia*, 100% ML, 1 PP; *N. kunmingense*, 97% in ML, 0.91 in PP; *N. rugosa*, 100% in ML, 0.94 in PP; and *N. simplicillium* 100% in ML and 0.97 in PP). Pairwise homoplasy index tests revealed no significant genetic recombination (*p* > 0.05) among the four closely related species groups. The homoplasy relationships among all species included in the phylogenetic tree are presented in Figure 2. Phylogenetic analysis (Figure 1) and homoplasy relationships (Figure 2) provide strong evidence for the genetic distinctiveness of the four new *Neocosmospora* species. According to the GCPSR species concept, four new species from these three groups were considered separate species, supporting the recognition of *N. hypertrophia*, *N. kunmingense*, *N. rugosa*, and *N. simplicillium*.

### 3.2. Taxonomy

*Neocosmospora hypertrophia* X. H. Wang, M. Jia & Y. B. Wang sp. nov. (Figure 3).

Index Fungorum no. IF903953.

*Etymology*: Named after the hypertrophic macroconidia.

Conidiophores erect, curved, or prostrate on substrate mycelia, or directly producing abundant conidia on aerial mycelia, straight or curved, smooth- and thin-walled, solitary or rarely branched, terminal, thickened at the base of phialides, (57.5–)59–70(–73) × 2.5–3.5(–4) μm (av. 67.5 × 3.1 μm (*n* = 17), sporulation site surrounded by an inconspicuous thickening and shortening, not extending beyond the collarette. Sporodochia are cream, gray, or gray-black. Microconidia abundant on CLA and PDA, but less on SNA, fusiform, claviform to subcylindrical, 0–1-septate, smooth- and thin-walled, (1.5–)2–3.5(–4) × (4–)5–8.5(–10) μm (av. 6.97 × 2.89 μm, *n* = 34), clustering in phialide forming false head arrangement; macroconidia abundant on CLA and SNA, but not on PDA, fusiform to falcate, slightly curved dorsal, gradually narrowed to both ends, papillate basal cells, blunt and curved apical cells, 1–3-septate, smooth- and thick-walled. 1-septate conidia: (16.5–)18.5–24.5 × (2.5–)3–3.5 μm (av. 21.6 × 3.4 μm, *n* = 17); 2-septate conidia: (24–)24.5–27(–27.5) × 3.5–4 μm (av. 26 × 3.8 μm, *n* = 21); 3-septate conidia: (28–)29–33(–36) × 4–5(–5.5) μm (av. 30.9 × 4.7 μm, *n* = 25). Chlamydospores were not observed.

*Culture characteristics*: Colonies on PDA were cultured at 25 °C in the dark, with a diameter of 5.65 cm at 7 d. The colonies were dry, white, silky, and thin in the middle, with a thick white ring around the edges; the reverse was white to straw. Colonies grown on OA in the dark at 25 °C for 7 d could reach 3.55 cm in diameter, white, cream, with abundant hyphae, a feathery appearance, and velvety texture; reverse straw.

*Material examined*: China, Yunnan Province, Pu’er City, isolated from the root of *D. cambodiana*, July 2020, M. Jia (Holotype HKAS 126203; ex-type culture YIM F00427); China, Yunnan Province, Pu’er City, isolated from the root of *D. cambodiana*, July 2020, M. Jia (cultures YIM F00490, YIM F00403, YIM F00422, YIM F00497).

*Notes*: The five isolates representing *N. hypertrophia* formed a strongly supported, genealogically exclusive lineage in the phylogeny inferred from the combined ITS, nrLSU, *rpb1*, *rpb2*, and *tef1* genes (Figure 1). *N. hypertrophia* is closely related to *N. pallidimors*, *N. simplicillium*, *N. stercicola*, *N. witzenhausenense* and *N. xiangyunense*, but differs by 13 bp from *N. pallidimors* in the three-gene (ITS-nrLSU-*tef1*) dataset (*rpb1* and *rpb2* genes are not available for *N. pallidimors*), 28 bp and 32 bp from *N. simplicillium* and *N. stercicola* in the five-gene (ITS-nrLSU-*tef1*-*rpb1*-*rpb2*) dataset, 32 bp from *N. witzenhausenense* in the four-gene (ITS-nrLSU-*tef1*-*rpb2*) dataset (*rpb1* gene is not available for *N. witzenhausenense*), and 6 bp from *N. xiangyunense* in the two-gene (ITS-*tef1*) dataset (nrLSU, *rpb1* and *rpb2* genes are not available for *N. xiangyunense*). Morphologically, *N. pallidimors*, *N. simplicillium*, and *N. stercicola* produce chlamydospores, whereas *N. hypertrophia* does not produce. In contrast to *N. stercicola*, which exhibits pale orange sporodochial coloration, and *N. witzenhausenense*, which has white sporodochia, *N. hypertrophia* is characterized by gray to gray-black sporodochia. Compared to *N. pallidimorum* and *N. simplicillium*, *N. hypertrophia* has wider macroconidia. Furthermore, the PHI test revealed no significant recombination (*p* = 0.5848) between *N. hypertrophia* and its closely related taxa (Figure 2). Therefore, *N. hypertrophia* is proposed as a novel species based on morphological and phylogenetic evidence.

*Neocosmospora kunmingense* X. H. Wang, M. Jia & Y. B. Wang sp. nov. (Figure 4).

Index Fungorum no. IF903954.

Etymology: Named after Kunming City, where the species was collected.

Abundant phialides arise from aerial mycelia, which are straight, smooth, thin-walled, simple, unbranched, and terminal. Phialides subcylindrical, (23.5–)25–35.5(–37) × (2.5–)3–4.5 μm (av. 30.2 × 3.8 μm, *n* = 19), sporulation site surrounded by an inconspicuous thickening and shortening, not spreading out of the collarette. Sporodochia cream, which is pale yellow to brown, rapidly develops into spores. Microconidia abundant on PDA and SNA, but not on CLA, fusiform, claviform to subcylindrical, straight or curved, 0–1(–2)-septate, smooth- and thin-walled, 6.5–18(–18.5) × (1–)2–3(–3.5) μm (av. 2.68 × 12.8 μm, *n* = 43), clustering at monophialides formed false head arrangement. Macroconidia were abundant on CLA and SNA but not on PDA, fusiform to falcate, slightly curved dorsally, gradually tapered at both ends, with inconspicuous papillate basal cells, blunt and curved apical cells, 1–3-separated, smooth thick walled; 1-septate conidia: (15–)16.5–24(–25) × (2–)2.5–3 μm (av. 21.9 × 2.7 μm, *n* = 3); 2-septate conidia: (23–)24.5–31.5(–32.5) × (3–)3.5–4(–4.5) μm (av. 29.3 μm, *n* = 14); 3-septate conidia: (30–)33 × 51(–52.5) × 4.5–5(–5.5) μm (av. 41 × 3.9 μm, *n* = 12). Chlamydospores not observed.

*Culture characteristics*: Colonies on PDA were cultured at 25 °C in the dark, with a diameter of 6.15 cm in 7 d; colonies were moist, white, conidial-like hyphae thinner in the middle, and formed a thick white ring at the margin; reverse was straw. Colonies cultured on OA were incubated at 25 °C for 7 d in the dark, reaching 4.75 cm in diameter, white, cream to pale yellow, with abundant mycelia resembling cashmere; reverse white to straw.

*Material examined*: China, Yunnan Province, Kunming County, isolated from the root of *D. cambodiana*, October 2019, M. Jia (Holotype HKAS 126204; ex-type culture YIM F00502); China, Yunnan Province, Kunming County, isolated from the leaf and root of *D. cambodiana*, October 2019, M. Jia (cultures YIM F00361, YIM F00315, YIM F00373).

*Notes*: Isolates representing *N. kunmingense* were resolved as a strongly supported genealogically exclusive lineage in the phylogeny inferred from the combined ITS, nrLSU, *rpb1*, *rpb2*, and *tef1* genes (Figure 1). *N. kunmingense* is closely related to *N. solani* and *N. rubicola*, but differs by 12 bp from *N. solani* in the five-gene (ITS-nrLSU-*tef1*-*rpb1*-*rpb2*) dataset and 21 bp from *N. rubicola* in the four-gene (ITS-nrLSU-*tef1*-*rpb2*) dataset (*rpb1* gene is not available for *N. rubicola*). Morphologically, *N. kunmingense* can be distinguished from *N. solani* and *N. rubicola* by its false-head arrangement of microconidia, fewer septa, smaller conidia, and the rare occurrence of mononematous conidiophores. In *N. solani* and *N. rubicola*, macroconidia typically possess more than three septa, whereas in *N. kunmingense*, macroconidia have a maximum of three septa. Furthermore, the PHI test showed no significant recombination (*p* = 0.1) between *N. kunmingense* and its closely related taxa (Figure 2). Thus, *N. kunmingense* is introduced as a new species.

*Neocosmospora rugosa* X. H. Wang, M. Jia & Y. B. Wang sp. nov. (Figure 5).

Index Fungorum no. IF903956.

Etymology: Named after the colony characteristics on PDA.

Conidiophores produced on aerial hyphae, smooth- and thin-walled, simple or 2-branched, thin rectangular, 25.5–34(–38) × (2.5–)3.5–4 μm (av. 29.3 × 3.4 μm, *n* = 20), sporulation site surrounded by an inconspicuous thickening and shortening, not extending beyond the collarette. Sporodochia are cream, yellow, or gray. Microconidia were abundant on PDA and SNA but not on CLA, fusiform, claviform, or obovoid, straight or curved, 0–1-septate, smooth- and thin-walled, (3.5–)5.5–7.5(–9.5) × 1.5–2.5 μm (av. 6.1 × 2.1 μm, *n* = 51), clustered in phialides forming a false head arrangement; macroconidia abundant on CLA and PDA, but not on SNA, fusiform to falcate, slightly curved dorsally, gradually tapered to both ends, with inconspicuous papillate basal cells, and blunt and curved apical cells, 3–5-septate, smooth- and thick-walled, (22–)29–47(–56) × 3–3.5(–4) μm (av. 28.3 × 2.5 μm, *n* = 17). Chlamydospores not observed.

*Culture characteristics*: Colonies on PDA were cultured at 25 °C in the dark, and the diameter reached 6.4 cm in 7 d. The colonies were moist, wrinkled, white, and conidial, with hyphae growing upward and thinner in the center, forming a thick white ring at the margin, and the reverse was straw to white. Colonies cultured on OA were incubated in the dark at 25 °C for 7 d, up to 3 cm in diameter, white, cream, with abundant aerial hyphae, central hyphae growing upward, panniform, and reverse straw to white.

*Material examined*: China, Yunnan Province, Pu’er City, isolated from the roots of *D. cambodiana*, October 2020, M. Jia (Holotype HKAS 135088; ex-type culture YIM F00493).

*Note*: The recognition of *N. rugosa* as a distinct species is strongly supported by both phylogenetic and morphological evidence, despite its description being based on a single isolate. In the concatenated analysis of ITS, LSU, *tef1*, *rpb1*, and *rpb2* genes, *N. rugosa* formed a highly supported clade (ML-BS = 100%, BI-PP = 0.94). *N. rugosa* is closely related to *N. longissima*, *N. oblonga*, *N. maoershanica*, and *N. paraeumartii*, but differs by 70 bp, 57 bp, and 69 bp from *N. longissima*, *N. oblonga* and *N. paraeumartii* in the five-gene (ITS-nrLSU-*tef1*-*rpb1*-*rpb2*) dataset, and 27 bp from *N. maoershanica* in the three-gene (ITS-*tef1*-*rpb2*) dataset (nrLSU and *rpb1* genes are not available for *N. maoershanica*). This level of sequence divergence surpasses the established species delineation thresholds within the genus *Neocosmospora*, particularly concerning the *tef1* and *rpb2* genes, which are considered critical markers for this genus [21]. Morphologically, *N. rugosa* exhibits unique characteristics that distinguish it from related species (*N. longissima*, *N. oblonga*, *N. maoershanica*, and *N. paraeumartii*). These include its 3–5-septate macroconidia (compared to 1–3-septate in *N. oblonga*), yellow-gray sporodochia (in contrast to the white sporodochia observed in *N. longissima*), and a unique wrinkled colony morphology on PDA medium (Figure 6a). These phenotypic discontinuities are consistent with the Genealogical Concordance Phylogenetic Species Recognition (GCPSR) criterion [46]. Although *N. rugosa* is described from a single isolate, limiting our ability to assess intraspecific genetic and phenotypic variation, the formal description of fungal species based on unique isolates is well-established and accepted practice-particularly when supported by robust phylogenetic placement and distinct morphological traits [21]. The distinct phylogenetic lineage and morphological feature collectively support the establishment of *N. rugosa* as a novel species.

*Neocosmospora simplicillium* X. H. Wang, M. Jia & Y. B. Wang sp. nov. (Figure 6).

Index Fungorum no. IF903955.

*Etymology*: Named after its single phialide.

Conidiophores produced on aerial hyphae, smooth- and thin-walled, simple, and unbranched. Phialides nearly cylindrical to bottle-shaped, (51.5–)58.5–73.5(–82) × 3–4.5(–5.5) μm (av. 65.7 × 3.9 μm, *n* = 11), porulation site surrounded by an inconspicuous thickening and shortening, not extending beyond the collarette. Microconidia were abundant on CLA and SNA but not on PDA, ovoid, claviform, or obovoid, 0–1-septate, smooth- and thin-walled, (5–)7 × 11(–13) μm (av. 9 × 1.1 μm, *n* = 31), clustering in monophialides forming a false head arrangement; macroconidia abundant on SNA and PDA, but not on CLA, fusiform to falcate, slightly curved dorsoventrally, gradually tapered to both ends, with inconspicuous papillate basal cells, and blunt, curved apical cells, 1–3-septate, smooth- and thick-walled; 1-septate conidia: (13.5–)16–21(2) × (2.5–)3–4(–5) μm (av. 18.6 × 3.1 μm, *n* = 15); 2-septate conidia: (19–)21.5–26(–27.5) × (3–)3.5–4(–4.5) μm (av. 23.2 × 4 μm, *n* = 13); 2-septate conidia: 26–31.5(–34) × 3.5–4.5(–5.5) μm (av. 29 × 4.3 μm, *n* = 10); chlamydospores abundant, globose to subglobose, smooth or rough, thick-walled, (6–)6.5–8.5(–9) × (5.5–)6–7(–7.5) μm (av. 7.7 × 6.5 μm, *n* = 18), terminal or intercalary, solitary.

*Culture characteristics*: Colonies on PDA were cultured at 25 °C in the dark, with a diameter of 6.25 cm at 7 d. The colony surface was moist, pale yellow to white, panniform, with abundant aerial at the margin, and the margin was entire. Reverse straw to white. Colonies cultured on OA were incubated at 25 °C for 7 d in the dark, with a diameter of 4.20 cm, colony surface rough, white to light orange, with a distinct villous structure in the center, margin with light orange aerial hyphae, and reverse straw to amber.

*Material examined*: China, Yunnan Province, Pu’er City, isolated from the root of *D. cambodiana*, October 2020, M. Jia (Holotype HKAS 135087; ex-type culture YIM F00563); China, Yunnan Province, Pu’er City, isolated from the root of *D. cambodiana*, October 2020, M. Jia (cultures YIM F00656, YIM F00566).

*Notes*: Isolates representing *N. simplicillium* were resolved as a strongly supported genealogically exclusive lineage in the phylogenies inferred from the combined ITS, nrLSU, *rpb1*, *rpb2*, and *tef1* genes (Figure 1). *N. simplicillium* is closely related to *N. hypertrophia*, *N. pallidimors*, *N. stercicola*, *N. witzenhausenense* and *N. xiangyunense*, but differs by 28 bp and 22 bp from *N. hypertrophia* and *N. stercicola* in the five-gene (ITS-nrLSU-*tef1*-*rpb1*-*rpb2*) dataset, 9 bp from *N. pallidimors* in the three-gene (ITS-nrLSU-*tef1*) dataset (*rpb1* and *rpb2* genes are not available for *N. pallidimors*), 17 bp from *N. witzenhausenense* in the four-gene (ITS-nrLSU-*tef1*-*rpb2*) dataset (*rpb1* gene is not available for *N. witzenhausenense*), and 5 bp from *N. xiangyunense* in the two-gene (ITS-*tef1*) dataset (nrLSU, *rpb1* and *rpb2* genes are not available for *N. xiangyunense*). Morphologically, *N. simplicillium* can be distinguished from its closest relatives by the presence of abundant woolly aerial hyphae, in contrast to the sparse and flocculent hyphae observed in *N. hypertrophia*, *N. pallidimors*, *N. stercicola*, and *N. witzenhausenense*. Additionally, while related species possess branched phialides, *N. simplicillium* is characterized by simple, unbranched phialides. Furthermore, the PHI test revealed no significant recombination (*p* = 0.5848) between *N. simplicillium* and its closely related taxa (Figure 2). Therefore, based on both molecular phylogenetic and morphological evidence, *N. simplicillium* is proposed as a novel species.

## 4. Discussion

The taxonomic relationship between *Fusarium* and *Neocosmospora* has been controversial owing to the conflict between taxonomy and practical considerations, particularly in medical and agricultural applications. This debate especially focused on whether species in the *F. solani* species complex (FSSC) should be replaced from *Fusarium* to *Neocosmospora*. Several scholars have contended that the traditional morphological concept of *Fusarium* is polyphyletic based on phylogenetic evidence and have proposed dividing the genus into several lineages [47,48,49]. They supported a more precise concept of *Fusarium* (= *Gibberella*) to include groups at the F3 monophyletic node. Under this revised framework, several lineages previously included in *Fusarium*, including the FSSC, have been reassigned to *Neocosmospora* and other genera [6,21,47,48,49,50,51]. Conversely, some researchers believe that maintaining the broad concept of *Fusarium* is essential for consistency in clinical and agricultural contexts, where the name ‘*Fusarium*’ is deeply entrenched in disease reporting and management practices [52]. They argue that retaining FSSC within *Fusarium* aids in communication and intervention strategies related to *Fusarium* in plant and human health [52,53]. Crous et al. proposed a comprehensive reclassification of the *Fusarium* complex into ten distinct genera based on an integrative approach combining morphology, phylogeny, secondary metabolite profiles, and ecology [21]. Given the robustness of this classification system and its alignment with evolutionary relationships, we adopted the nomenclature proposed by Crous et al. in the present study [21].

In this study, four novel species i.e., *N. kunmingense*, *N. hypertrophia*, *N. rugosa*, and *N. simplicillium* were isolated from *Dracaena* plants in Yunnan Province, China, and identified using an integrative taxonomic approach based on molecular phylogeny and morphological characteristics. This discovery highlights the likely underestimation of fungal diversity associated with this host. Four novel taxa were identified, suggesting that *Dracaena*-associated *Neocosmospora* may harbor greater species richness than previously recognized. The phylogenetic placement of these newly described *Neocosmospora* species within a clade that includes known plant and mammalian pathogens indicates their potential pathogenic capability [16,54,55]. However, no pathogenicity assays or field observations were conducted in this study to confirm whether they were pathogenic, and their ecological functions remain largely unknown. These findings underscore the need for further investigation into the biological roles, host range, and potential pathogenicity of these newly identified species, particularly in relation to their ecological associations with *Dracaena*. Understanding the interaction between *Dracaena* and *Neocosmospora* could yield significant insights into host-fungus dynamics, with implications for both ecological studies and practical applications in agriculture and biotechnology. Several *Neocosmospora* species have demonstrated the capacity to produce bioactive secondary metabolites, including antimicrobial and cytotoxic compounds, suggesting their potential for pharmaceutical or industrial exploitation [56]. Therefore, continued research on *Dracaena*-associated *Neocosmospora* species is not only essential for understanding of fungal biodiversity but also for uncovering novel biological resources.

Despite the phylogenetic support provided by multiple genes (ITS, nrLSU, *tef1*, *rpb1*, and *rpb2*), the current phylogenetic tree within *Neocosmospora* still exhibits short branch lengths and low bootstrap values at certain nodes, reflecting limited genetic divergence among closely related species [57,58,59]. Future studies should prioritize whole-genome sequencing and population-level analyses to better understand the genetic structure, geographic distribution, and evolutionary history of *Neocosmospora* species. Expanding sampling efforts across Southeast Asia and beyond will also aid in clarifying the biogeographic patterns and ecological roles of this increasingly recognized fungal genus.

## Figures and Tables

**Figure 1 jof-11-00571-f001:**
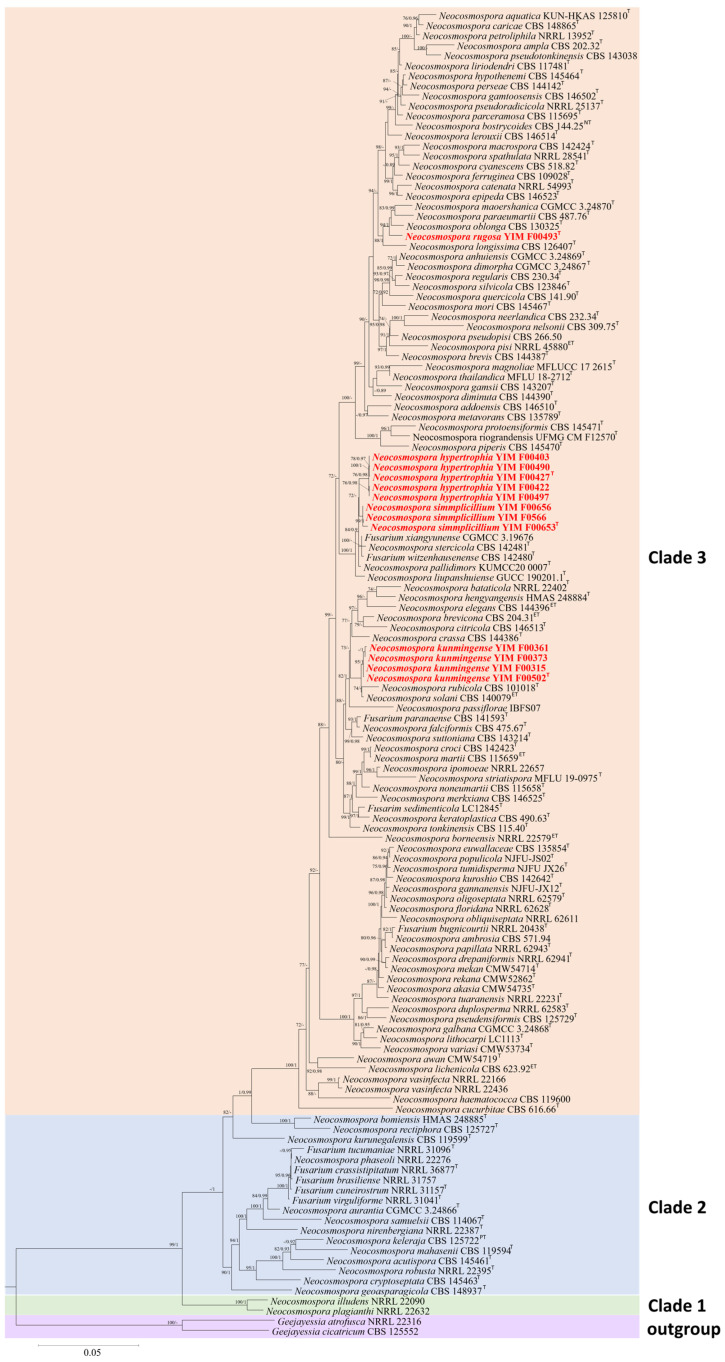
Phylogenetic tree inferred from the combined ITS-nrLSU-*rpb1*-*rpb2*-*tef1* genes of *Neocosmospora* species. *Geejayesia atrofusca* (NRRL 22316) and *G. cicatricum* (CBS 125552) are designated as the outgroup. The RAxML Bootstrap support values (ML-BS > 70%) and Bayesian posterior probabilities (BI-PP > 0.9) are shown at the nodes (ML-BS/BI-PP). Ex-type, ex-epitype, and ex-neotype strains are indicated with T, ET, and NT, respectively.

**Figure 2 jof-11-00571-f002:**
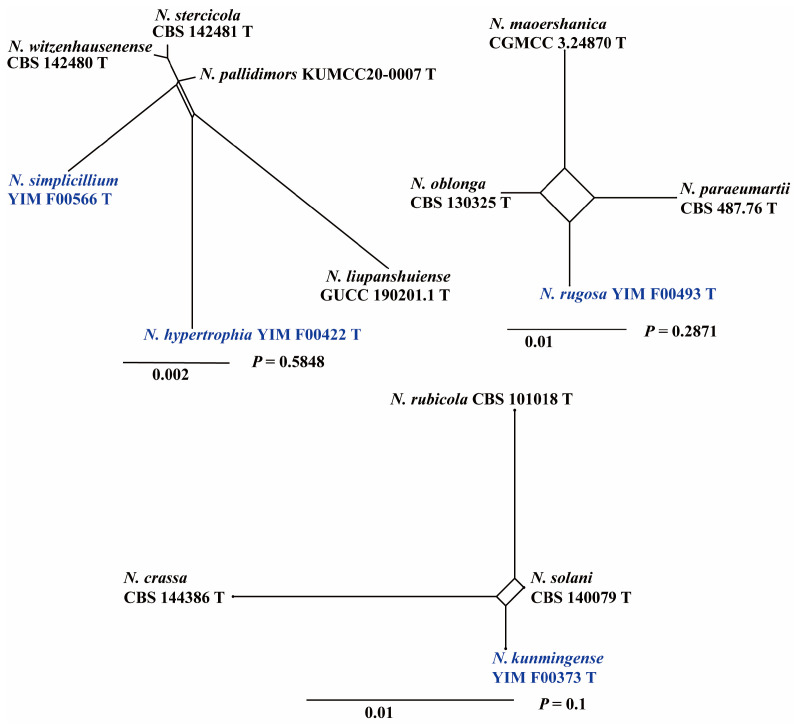
The pairwise homoplasy index (PHI) test of four new *Neocosmospora* species and their closely related species. New taxa are printed in bold blue.

**Figure 3 jof-11-00571-f003:**
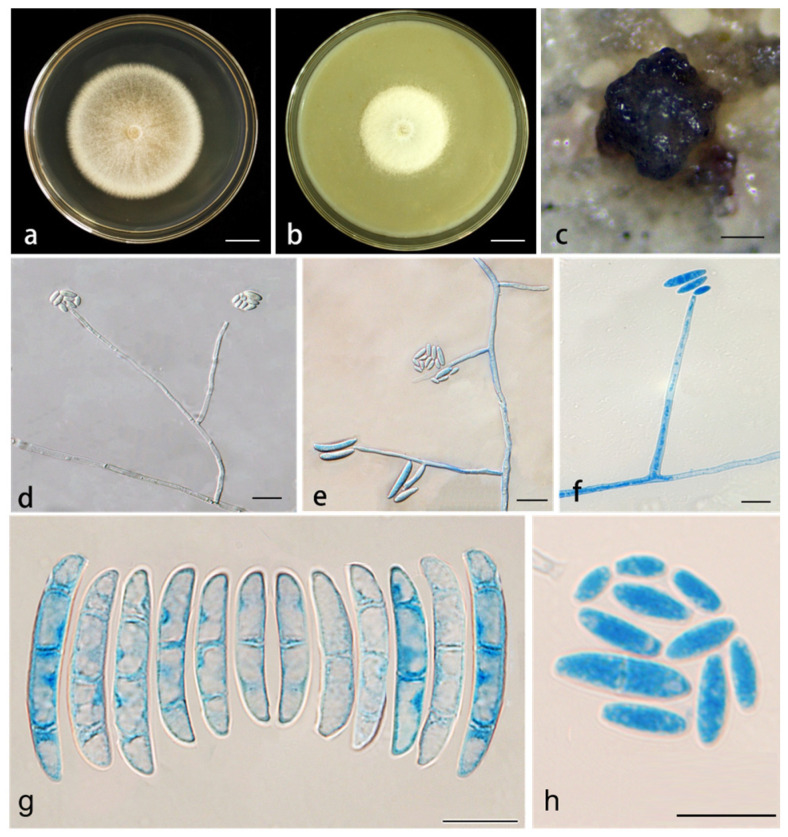
*Neocosmospora hypertrophia* (ex-type culture YIM F00427). (**a**,**b**) Colonies on PDA and OA, respectively, after 7 d at 25 °C in the dark; (**c**) sporodochia formed on the surface of carnation leaves; (**d**–**f**). aerial conidiophores and phialides; (**g**) macroconidia; (**h**) microconidia. Scale bars: (**a**,**b**) = 1.5 cm; (**c**) = 200 µm; (**d**–**f**) = 10 µm; (**g**,**h**) = 20 µm.

**Figure 4 jof-11-00571-f004:**
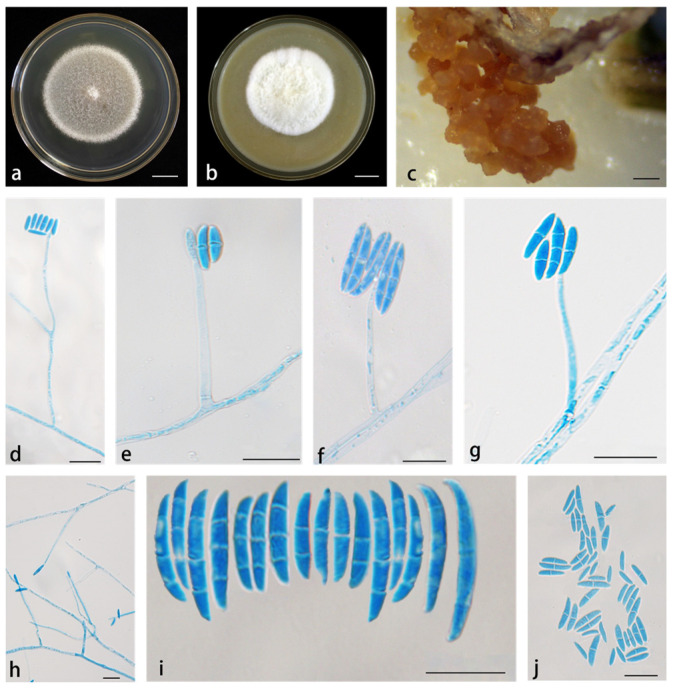
*Neocosmospora kunmingense* (ex-type culture YIM F00502). (**a**,**b**) Colonies on PDA and OA, respectively, after 7 d at 25 °C in the dark; (**c**) sporodochia formed on the surface of carnation leaves; (**d**–**h**) aerial conidiophores and phialides; (**i**) macroconidia; (**j**) microconidia. Scale bars: (**a**,**b**) = 1.5 cm; (**c**) = 200 µm; (**d**–**h**) = 10 µm; (**i**,**j**) = 20 µm.

**Figure 5 jof-11-00571-f005:**
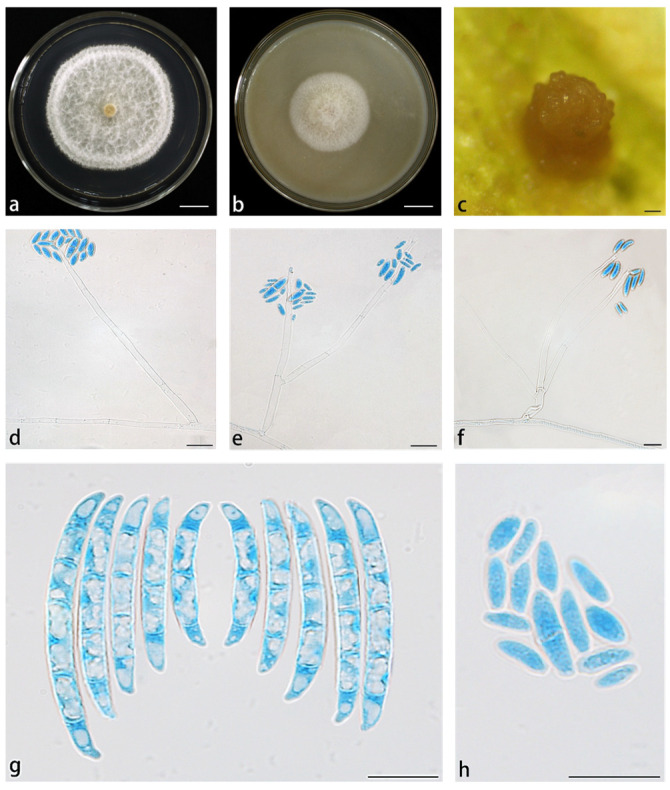
*Neocosmospora rugosa* (ex-type culture YIM F00493). (**a**,**b**) Colonies on PDA and OA, respectively, after 7 d at 25 °C in the dark; (**c**) sporodochia formed on the surface of carnation leaves; (**d**–**f**) aerial conidiophores and phialides; (**g**) macroconidia; (**h**) microconidia. Scale bars: (**a**,**b**) = 1.5 cm; (**c**) = 200 µm; (**d**–**f**) = 10 µm; (**g**,**h**) = 20 µm.

**Figure 6 jof-11-00571-f006:**
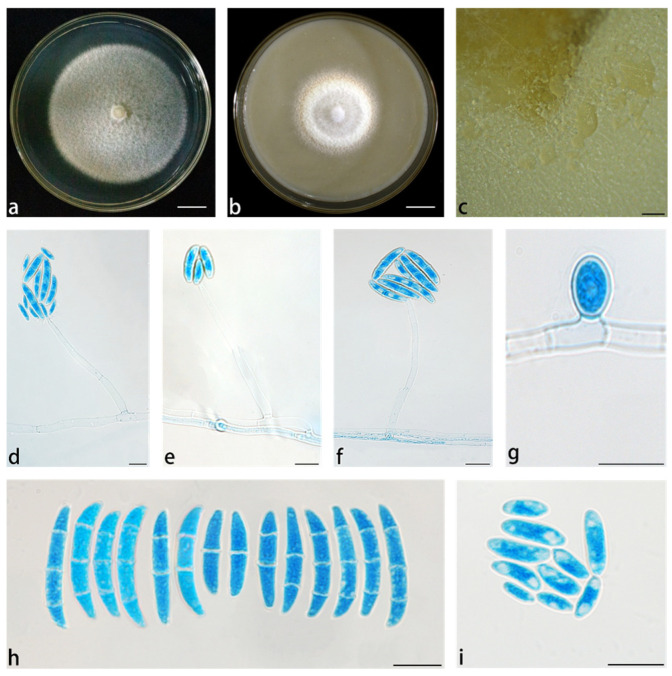
*Neocosmospora simplicillium* (ex-type culture YIM F00563). (**a**,**b**) Colonies on PDA and OA, respectively, after 7 d at 25 °C in the dark; (**c**) sporodochia formed on the surface of carnation leaves; (**d**–**f**) aerial conidiophores and phialides; (**g**) chlamydospores; (**h**) macroconidia; (**i**) microconidia. Scale bars: (**a**,**b**) = 1.5 cm; (**c**) = 200 µm; (**d**–**g**) = 10 µm; (**h**,**i**) = 20 µm.

**Table 1 jof-11-00571-t001:** PCR primers for DNA amplification of *Neocosmospora* in this study.

Gene	Primer	5′–3′ Sequence	Annealing Temperature (°C)	Reference
ITS	ITS1	TCCGTAGGTGAACCTGCGG	55	[29]
	ITS4	TCCTCCGCTTATTGATATGC		
nrLSU	LR5	ATCCTGAGGGAAACTTC	52	[30]
	LR0R	GTACCCGCTGAACTTAAGC		
*tef1*	EF1	ATGGGTAAGGARGACAAGAC	57	[31]
	EF2	GGARGTACCAGTSATCATG		
*rpb1*	*RPB1*-Fa	CAYAARGARTCYATGATGGGWC	58 (5 cycles) → 57 (5) → 56 (35)	[32]
	*RPB1*-G2R	GTCATYTGDGTDGCDGGYTCDCC		
*rpb2*	*RPB2*-5f2	GGGGWGAYCAGAAGAAGGC	59	[33]
	*RPB2*-7cr	CCCATRGCTTGYTTRCCCAT		
	*RPB2*-7cf	ATGGGYAARCAAGCYATGGG	58	[34]
	*RPB2*-11ar	GCRTGGATCTTRTCRTCSACC		

## Data Availability

All the sequences in this study have been deposited in GenBank (http://www.ncbi.nlm.nih.gov, accessed on 14 December 2023) (see Appendix A for the accession numbers).

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
