# Peer review of "Molecular and Morphological Evidence Reveals Four New Neocosmospora Species from Dragon Trees in Yunnan Province, China"

_jof, 2025, doi:10.3390/jof11080571_

Round 1

Reviewer 1 Report

The manuscript is well-written and easy to follow. However, the relevance of the study is not well described in the introduction, so it should be improved. More details about the sampling approach used should be added to the methodology section. Results are well described and follow a logical structure. Discussion could be improved to better illustrate the contribution of the study to the field. I have provided a few suggestions to enhance the clarity of the figures and strengthen the contextual background.
I look forward to seeing the revised version.

Abstract

- Consider adding a closing statement to the abstract that highlights the relevance and significance of the study to broader scientific or practical contexts.

Introduction

- The relationship between Neocosmospora and Dracaena is not clearly explained. Was Neocosmospora previously detected in Dracaena, or is this association a novel finding of the study? Clarifying this point would help readers better understand the rationale behind the research.
- I recommend revising the introduction to clearly articulate the study’s objectives and justify the importance of describing Neocosmospora in association with Dracaena.

Materials and Methods

- The sampling strategy is unclear. Please specify how the plants were selected and what criteria were used.
- I suggest moving Table 2 to the supplementary materials, as it may not be essential to the main text.
- Consider relocating Figure 1 to the Results section (preferably after or alongside section 3.1), and ensure the findings it illustrates are properly discussed.

Results

- Section 3.1 would benefit from a more structured opening. Start by stating the total number of fungal strains isolated, followed by the number of species identified, and then emphasize that four of them are potential new species.
- Review and correct the italicization of species names throughout the manuscript for consistency.
- Line 191: Please include the bootstrap (BS) and posterior probability (PP) values that support the distinct phylogenetic clade, as this information is important for evaluating the robustness of your results.
- Figures: It is good practice to present images in the following order—fungal structures in host, followed by culture plates, then microscopy images. I recommend reordering the figure panels accordingly. Also, make sure that each image includes a clearly indicated scale bar with its measurement in the image, not in the legend.
- Are MycoBank numbers available for each proposed new species? If so, please include them in the manuscript.

Reviewer 2 Report

The manuscript entitled “Molecular and Morphological Evidence Reveals Four New Neocosmospora Species from Dragon’s Blood Tree in Southwestern China” demonstrates a mycological research focused on the fusarioid taxa associated with the Dragon’s Blood tree species Dracaena cambodiana and D. louri.

Based on morphological and phylogenetic evidence, the authors introduce four new species, namely Neocosmospora kunmingense, N. hypertrophia, N. simplicillium, and N. wrinkles.

The manuscript is well-written, and the novel species are properly described with the necessary information provided.

However, I have a suggestion regarding the methodology mentioned in paragraph 2.1.

The procedure followed by the authors to obtain endophytic fungi does not guarantee this scope. This procedure doesn’t exclude the plethora of phylloplane and rhizosphere microorganisms. Thus, I suggest excluding the term endophytic from the text, since it is not properly supported by the methodology.
